# Role of Daptomycin in Cutaneous Wound Healing: A Narrative Review

**DOI:** 10.3390/antibiotics11070944

**Published:** 2022-07-14

**Authors:** Giulio Rizzetto, Elisa Molinelli, Giulia Radi, Federico Diotallevi, Oscar Cirioni, Lucia Brescini, Andrea Giacometti, Annamaria Offidani, Oriana Simonetti

**Affiliations:** 1Clinic of Dermatology, Department of Clinical and Molecular Sciences, Polytechnic University of Marche, 60126 Ancona, Italy; grizzetto92@hotmail.com (G.R.); molinelli.elisa@gmail.com (E.M.); radigiu1@gmail.com (G.R.); federico.diotallevi@hotmail.it (F.D.); a.offidani@ospedaliriuniti.marche.it (A.O.); 2Clinic of Infectious Diseases, Department of Biomedical Sciences and Public Health, Polytechnic University of Marche, 60121 Ancona, Italy; o.cirioni@staff.univpm.it (O.C.); l.brescini@staff.univpm.it (L.B.); a.giacometti@staff.univpm.it (A.G.)

**Keywords:** daptomycin, wound healing, staphylococcal skin infection, Vitamin E, IB-367, RNA III-inhibiting peptide

## Abstract

Daptomycin is active against Gram-positive bacteria, including methicillin-resistant *Staphylococcus aureus* (MRSA) and the on-label indications for its use include complicated skin and skin structure infections (cSSSI). We performed a narrative review of the literature with the aim to evaluate the role of daptomycin in the skin wound healing process, proposing our point of view on the possible association with other molecules that could improve the skin healing process. Daptomycin may improve wound healing in MRSA-infected burns, surgical wounds, and diabetic feet, but further studies in humans with histological examination are needed. In the future, the combination of daptomycin with other molecules with synergistic action, such as vitamin E and derivates, IB-367, RNA III-inhibiting peptide (RIP), and palladium nanoflowers, may help to improve wound healing and overcome forms of antibiotic resistance.

## 1. Introduction

Daptomycin is a cyclic lipopeptide that is active against Gram-positive bacteria, including vancomycin-resistant *Enterococci* (VRE) and methicillin-resistant *Staphylococcus aureus* (MRSA) [1,2]. The on-label indications include, in addition to bacteremia and endocarditis, complicated skin and skin structure infections (cSSSI) [2,3].

The mechanism of action of daptomycin disrupts the functions of the Gram-positive bacterial cell membrane, causing membrane depolarization through ion leakage [4]. Daptomycin forms a unique complex with calcium ions and phosphatidylglycerol molecules in bacterial membranes which are supposed to allow the formation of transient ionophores and lead to bacterial death by intracellular inhibition of DNA and RNA [4,5,6]. This occurs since adenosine tri-phosphate (ATP) regeneration through oxidative phosphorylation stops when the membrane potential collapses. In addition, the loss of cellular Mg^2+^, which normally stabilizes ATP, should contribute to ATP depletion, which impairs macromolecular biosynthesis, including the DNA, RNA, protein, and peptidoglycan synthesis pathways [6].

Recent evidence suggests an action on actively dividing regions of bacterial membranes, causing a re-organization in the distribution of membrane lipids and proteins, which secondarily results in membrane leakage and depolarization [7,8,9].

In terms of the pharmacokinetics, the gastrointestinal tract is not able to absorb daptomycin, for this reason intravenous administration is recommended. Since daptomycin binds with plasma proteins to 90% and has a negative charge at a physiological pH, the volume of distribution is limited, 0.1 l/Kg in healthy subjects. Elimination is predominantly renal, with 50% being excreted unmodified in the urine. This explains the need for dosage adjustment in patients with renal insufficiency [10,11].

However, emerging evidence of MRSA that is resistant to daptomycin after repetitive antibiotic exposure suggests that we should evaluate the combination of daptomycin with other molecules both to overcome possible bacterial resistance and to achieve a synergistic action, improving wound healing [12,13,14].

Most SSTIs in adults are caused by staphylococci, including MRSA, or β-haemolytic streptococci [15]. Vancomycin is the antibiotic of first choice for SSTIs when β-lactams are contraindicated or ineffective [16]. However, the increasing prevalence of resistance to vancomycin requires other options of treatment. Among these, daptomycin is a valid alternative that is approved for the treatment of cSSSIs (4 mg/kg die for 7–14 days) [17], and various meta-analyses confirm that there are no significant differences in the efficacy of daptomycin compared to standard drugs [18,19,20].

CSSSIs are also common in children [21,22,23,24], and guidelines recommend vancomycin as parenteral therapy or clindamycin as an alternative [25,26,27]. However, vancomycin requires monitoring of its blood concentration and renal function [25,28,29,30]. Clindamycin may be ineffective as empirical therapy against MRSA when the resistance rate exceeds 10% [25,26,27]. Linezolid is also a viable alternative, although, especially with treatments that are longer than 28 days, it may be associated with the risk of neurotoxicity and myelosuppression [31,32,33,34]. In a recent randomized trial [35], daptomycin was not inferior to vancomycin and clindamycin in pediatric patients, with a comparable safety profile, as confirmed in the study by Iwata et al. [2]. However, the clinician must be careful when using daptomycin, considering the risk of side effects such as acute eosinophilic pneumonia and increased creatine phosphokinase [20,36].

To the best of our knowledge, there are no reviews in the literature that provide an overview of the action of daptomycin in wound healing. For this reason, we performed a review of the literature to evaluate the role of daptomycin in cutaneous wound healing. (Table 1)

## 2. Results and Discussion

### 2.1. Burns

The role of daptomycin in wound healing was evaluated in burns. Burn patients requiring hospitalization are often at a higher risk of MRSA infection and appropriate antibiotic therapy must be administered [49,50,51]. In addition, MRSA can produce soluble products that can induce apoptosis of fibroblasts and prevent their proper organization in the tissue repair process [37].

In a murine model with MRSA-infected burn wounds [38], daptomycin was found to have a greater antimicrobial effect than teicoplanin. This can be explained in part by the greater penetration of daptomycin into soft tissues, with 70% of the plasma concentration found in the tissues that were examined [52]. Interestingly, a higher dose of 4 mg/kg may be more effective in patients with cSSSI as daptomycin has a dose-dependent bactericidal action and 7 mg/kg may help both to maximize the bactericidal action and avoid the development of bacterial resistance [38]. A recent study of Chinese patients with severe burns and MRSA infection also found that high doses of daptomycin (from 6 mg/kg/day up to 12 mg/kg/day) were more effective than the standard dose [53]. Even if this dosage is higher than what is recommended, no significant differences of endogenous creatinine clearance rate (Ccr), blood urea nitrogen (BUN), and total protein (TP) were found in patients before and 7 days after high-dose daptomycin treatment. The mean concentration of albumin on day 7 after high-dose daptomycin treatment was significantly higher than the baseline value (40.5 ± 4.6 vs. 35.0 ± 6.5, respectively, *p* = 0.006). [53] We recommend that the physician should always assess the risk-benefit ratio in the individual patient before considering high doses of daptomycin.

In addition, the group of mice that were treated with daptomycin showed improved wound healing with better collagen organization and faster re-epithelization. After 21 days, the skin repair in this group was similar to that of the group with a non-infected burn. Immunohistochemically, it was also seen that in the daptomycin group there was higher expression of wound healing markers such as epidermal growth factor receptor (EGFR) and fibroblast growth factor (FGF-2) [38].

Skin wound healing is a complex, multifactorial process involving inflammation, cell proliferation and migration, extracellular matrix deposition, re-epithelialization, angiogenesis, and remodeling. [54,55] In particular, growth factors, such as FGF-2 and epidermal growth factor (EGF), play key roles in fibroblast and keratinocyte proliferation [56]. The increased expression of EGFR accelerates re-epithelialization, angiogenesis, keratinocyte proliferation, and migration from the wound edges [57,58,59].

FGF-2 is a polypeptide that is involved in wound healing by regulating the synthesis and degradation of the extracellular matrix and stimulates the formation of granulation tissue [60,61,62,63]. It has been shown that exogenous FGF-2 can be used to treat keloids and hypertrophic scars [64,65,66]. This explains the results of daptomycin in murine wound healing of MRSA-infected burns with improved scar quality and well-organized, non-hypertrophic re-epithelialization [38].

Finally, a review assessed the efficacy of daptomycin in treating infected wounds and SSSIs, including burns (n = 134), with an eradication of infection in 98.5% of cases, but lacking a histological assessment of wound healing [39].

### 2.2. Surgical Site Infections

Popov et al. reported [40], in an observational study on 23 patients, the efficacy of daptomycin in the wound healing of deep sternal wound infection (DSWI) that was sustained by *S. aureus* after cardiac surgery. All the patients resolved the infection and achieved complete wound healing after 22 ± 13.4 days. Again, there was a lack of histological data specifically confirming the action of daptomycin.

Silvestri et al. reported a murine model with skin wounds that were infected with strains of *S. aureus* and *Enterococcus* spp. that were taken from patients’ surgical wounds [41]. The combination of tigecycline with rifampicin and daptomycin has a synergic bactericidal action compared to single treatment, confirming that the combination with daptomycin and other antibiotics can be very effective and overcome potential problems of multi-resistant species. Chamberlain et al. also demonstrated the efficacy of daptomycin in the treatment of chirurgic wounds that were infected with *S. aureus*, MRSA, and enterococci even when previous antibiotic therapy, including vancomycin, failed. However, evaluations on wound healing are lacking [67].

In our opinion, the effects of daptomycin in surgical site-related SSSIs may potentially be optimal, allowing better wound healing.

### 2.3. Diabetic Foot

In a pilot study on eight patients with MRSA-infected diabetic feet Ambrosch et al. evaluated the effect of daptomycin in both eradicating the infection and wound healing [42]. The concentration of interleukin-6 (IL-6), matrix metalloproteinase-9 (MMP-9), and metallopeptidase inhibitor 1 (TIMP-1) was assessed from wound secretions. Local IL-6 decreased in the first three days of therapy, MMP-9 decreased in the following days, and TIMP-1 increased, with a reduction in wound size and eradication of MRSA at the end of the 14-day therapy.

An increase in TIMP-1, an anti-protease, and a reduction in MMP-9, a protease, was seen before wound reduction on day 14. The latter is the predominant collagenase in normal wound healing, but, if over-expressed, may be associated with chronic ulcers that do not heal due to excessive collagenase activity [68,69,70,71,72,73]. This may also result in an imbalance with TIMP-1, which is reduced.

A limitation of this study is the lack of histological assessment of wound healing, although this is difficult in the diabetic foot. Wound fluid analysis is a good indicator of IL-6, MMP-9, and TIMP-1 expression [74,75,76,77].

We believe that daptomycin may be an excellent therapeutic option in patients with MRSA-infected diabetic foot or diabetic ulcers with difficult wound healing.

### 2.4. Daptomycin in Association with Other Molecules

#### 2.4.1. Daptomycin (Dap) Micelles-Stabilized Palladium Nanoflowers (Dap-PdNFs)

A new method consists of including daptomycin in micelles-stabilized palladium nanoflowers by exploiting the photothermal action to convert laser light (808nm) into thermal energy [43]. This method (Dap-PdNFs + laser group) was proven to be effective in vitro against *S. Aureus*, reducing its viability after 10 min of laser to 30.93% (vs. 79.76%, Dap + laser group, vs. 85.14% PdNPs + laser group). The effect on wound healing was studied on diabetic mice with *S. aureus*-infected wounds. The mice were then randomly divided into five groups: saline group, saline + laser irradiation (1.75 W/cm^2^ for 5 min), Dap-PdNFs (topical application), and Dap-PdNFs + laser irradiation. After 14 days the Dap-PdNFs + laser group showed the best reduction in wound area, 6.5%, vs. the laser alone group, 34.04%, vs. Dap + laser group, 25%, and vs. PdNPs + laser group, 17.36%. A histological examination of the main organs (heart, liver, spleen, lung, and kidney) was performed, showing no alterations, thus confirming the good biocompatibility of the method. In conclusion, Dap-PdNFs irradiated with 808 nm light was effective against bacteria around the wound, prevented wound infection, promoted local angiogenesis and epithelial tissue growth, and accelerated wound healing in diabetic mice. This may be an interesting method that deserves to be evaluated with larger sample size studies, considering the topical application and the good tolerability.

#### 2.4.2. Vitamin E and Derivates

Tocotrienols (T3s) is a vitamin E isomer with immunomodulatory activity. In a study by Pierpaoli et al., the action of T3s in combination with daptomycin was evaluated in a mouse model with MRSA-infected wounds [44]. The T3s and daptomycin combination group showed the best antibacterial action compared to the single treatment or placebo, and the markers of wound healing were also better (fibronectin type III expression and IL-24 mRNAs). Again, the histological aspects of wound healing are lacking, but this study suggests that the combination of daptomycin with T3s may further improve the tissue repair process.

Vitamin E (VE) is a known immunomodulator and was observed in a study by Provinciali et al. to play a role as an enhancer of the antimicrobial activity of both daptomycin and tigecycline in a murine model with MRSA-infected wounds [45]. The group with VE pre-treatment and daptomycin showed the highest antibacterial efficacy. An assessment of wound healing is lacking here, but this study could be a starting point for evaluating VE supplementation in those patients with a high risk of MRSA-infected ulcers or wounds, in order to enhance the action of daptomycin and to benefit from VE possible effect on wound healing [78,79,80].

In addition, an interesting aspect of daptomycin is its possible immunomodulatory action. In an in vitro model on mononuclear cells from peripheral blood, *S. aureus* stimulates the production of several inflammatory cytokines, such as IL-1β, IL-6, IL-8, interferon-γ (INF-γ), and tumor necrosis factor alpha (TNF-α). Daptomycin suppresses the levels of these proinflammatory cytokines in vitro, independently of dose, but in vivo this does not seem to be confirmed [81]. However, if VE was provided prior to wound infection with MRSA, the animals showed a significant increase in CD49b+ lymphocytes after daptomycin treatment, whereas this did not occur with daptomycin alone, suggesting its role of immunomodulator enhancer [81].

#### 2.4.3. IB-367

IB-367 is a synthetic protegrin (RGGLCYCRGRFCVCVGRCONH2) with both fungicidal [82] and bactericidal [83] activity. In a murine model with MRSA-infected wounds, the combination of IB-367 with daptomycin was the most effective in bactericidal activity compared to single therapies and the combination of IB-367 with teicoplanin. The antimicrobial effect of IB-367 was associated with increased cytotoxicity of Natural Killer (NK), without a higher cell count. IB-367 correlated with increased CD11b and Gr-1 cells 3 days after MRSA challenge, whereas both of these leukocyte populations were reduced at 10 days after. Again, there is no histological reference to wound healing, but this is an excellent example of how the combination of daptomycin with new peptides can help overcome bacterial resistance [46].

#### 2.4.4. RNA III-Inhibiting Peptide

RNA III-inhibiting peptide (RIP) is a seven-amino acid molecule that is capable of blocking RNA III synthesis. RNA III is a transcriptional unit of the staphylococcal accessory gene regulator (Agr) system and can regulate bacterial quorum sensing, as well as biofilm formation [84].

This molecule, topically applied at the concentration of 1 mg/mL, was used in combination with daptomycin, 6 mg/kg intravenously, in a subject with a diabetic foot ulcer, achieving complete wound closure in 24 weeks and above all avoiding amputation [47]. This is an excellent example of the potential of combining new topical molecules with systemic therapy with daptomycin, exploiting not only the antibacterial action but also that of improving wound healing.

#### 2.4.5. Dap@Au/Ag Nanorods

Dong et al. show that a nanometric antimicrobial system Dap@Au/Ag nanorods (Dap@Au/Ag NRs), consisting of daptomycin bound to silver-coated gold nanorods, is effective against MRSA [48]. In vitro and in vivo models showed that after exposure to a laser at 808 nm, 0.8 W/cm^2^, 40–60 s, Dap@Au/Ag NRs increased its antibacterial activity with a good photothermal effect. In comparison to conventional photothermal therapy (PTT), laser irradiation that was administered at the early stage of infection with a controlled temperature below 47 °C, not only significantly inhibits MRSA, but also prevents a large area of wound ulceration and promotes wound healing without any thermal damage to the wound surface. Dap@Au/Ag NRs are a promising potential antimicrobial agent with a good effect on wound healing, providing a new strategy to overcome drug-resistant bacterial infections.

## 3. Materials and Methods

We performed a narrative review of the literature by searching the PubMed database for the following keywords: daptomycin, wound healing, burn, skin infections, SSSI, and skin and soft tissue infections. We selected articles showing the effect of daptomycin on wound healing first in humans and then in animal models, preferring those with histological evidence. In vitro studies have been reported for completeness. The aim of this review is to evaluate the role of daptomycin in the skin wound healing process, proposing our point of view on the possible association with other molecules that could improve the skin healing process. We did not consider time limits in the selection of the literature.

## 4. Conclusions

Some findings support that daptomycin is an important modulator of wound healing in MRSA-infected wounds, (Figure 1) allowing better organization of fibroblasts and keratinocytes, although human studies with histological evidence to confirm this are still required. In our opinion, in the future, the combination of daptomycin and other molecules with synergistic action may help to improve wound healing and overcome forms of antibiotic resistance, in particular the combination with topical RIP also appears to be promising in diabetic patients with MRSA-infected wounds [84].

A further issue emerging from this review is the possibility of using a high dose (HD) of daptomycin for the treatment of complicated skin and soft tissue infections (cSSTIs). In a study of 155 Chinese patients, 108 were treated with standard dose (SD) (4 mg/kg) and 47 with HD (≥6 mg/kg) and it was shown that the group with HD achieved clinical stabilization earlier (72.34% vs. 52.78%, *p* = 0.023), with an equal distribution of adverse effects in the two groups [85]. From our point of view, this may be a therapeutic option that the clinician should be aware of in the treatment of cSSSIs, always considering the patient’s clinical condition and his comorbidities, but further studies with histological assessment are necessary to evaluate the impact on wound healing.

Finally, Blasco et al. recently identified a new daptomycin analogue, kynomycin, which showed higher activity against MRSA and lower cytotoxicity than daptomycin [86]. This new cyclic lipopeptide could represent a further option for overcoming resistance to daptomycin, but its action on wound healing still needs to be assessed.

## Figures and Tables

**Figure 1 antibiotics-11-00944-f001:**
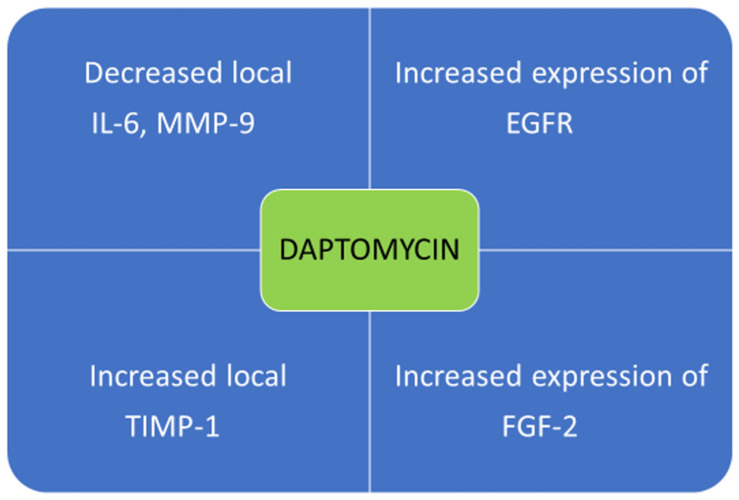
The possible immunomodulatory effects of daptomycin promoting cutaneous wound healing. Decreased local IL-6, MMP-9, increased local TIMP-1 [42] increased expression of EGFR, increased expression of FGF-2 [38] EGFR: epidermal growth factor receptor FGF-2: fibroblast growth factor-2 IL-6: interleukin-6. MMP-9: matrix metalloproteinase-9 TIMP-1: metallopeptidase inhibitor.

**Table 1 antibiotics-11-00944-t001:** Summary of the main evidence on the role of daptomycin in wound healing. EGFR: epidermal growth factor receptor FGF-2: fibroblast growth factor-2 IL-6: interleukin-6. MMP-9: matrix metalloproteinase-9 TIMP-1: metallopeptidase inhibitor.

	Main Evidence	References
**Burns**	Higher risk of MRSA infection in burn patients hospitalized MRSA-soluble products impair wound healing by apoptosis and disorganization of fibroblasts	Kirker et al. [37]
	Murine model, MRSA-infected burn wounds. Groups: one control not infected, one not infected + intraperitoneal daptomycin, one infected no treatment, one infected + intraperitoneal daptomycin, one infected + intraperitoneal teicoplanin. Daptomycin with greater antimicrobial effect than teicoplanin. (*p* < 0.01) Daptomycin groups with better epithelialization and significantly higher collagen scores, higher expression of EGFR and FGF-2 Daptomycin reduced hypertrophic burn scar formation.	Simonetti et al. [38]
	Infected wounds and SSSIs, including burns (134). Eradication of infection in 98.5% of cases Lack of histological assessment of wound healing	Friedman et al. [39]
**Surgical** **Site** **Infections**	23 patients deep sternal wound infection by *S. aureus* after cardiac surgery. All patients with complete wound healing after 22 ± 13.4 days. Lack of histological assessment of wound healing	Popov, A.F. et al. [40]
	Murine model, skin wounds infected with *S. aureus* and *Enterococcus* spp. Tigecycline + daptomycin with synergic bactericidal action compared to single treatment Lack of histological assessment of wound healing	Silvestri et al. [41]
**Diabetic Foot**	Pilot study, 8 patients with MRSA-infected diabetic feet treated with daptomycin IL-6 decreased in the first 3 days MMP-9 decreased TIMP-1 increased Reduction in wound size and eradication of MRSA after 14-day therapy Lack of histological assessment of wound healing	Ambrosch et al. [42]
	**Daptomycin in association with other molecules**	
**Daptomycin(Dap) micelles-stabilized palladium nanoflowers** **(Dap-PdNFs)**	In vitro, *S. Aureus* Dap-PdNFs + 808 nm laser *S. aureus* viability reduced after 10 min of laser to 30.93% In vivo, diabetic mice with *S. Aureus*-infected wounds. Five groups: saline group, saline + laser irradiation, Dap-PdNFs (topical application), Dap-PdNFs + laser irradiation. Dap-PdNFs + laser group showed the best reduction in wound area, 6.5%. Histological examination (heart, liver, spleen, lung, and kidney) with no alterations Dap-PdNFs + 808 nm light were effective against bacteria around the wound, promoted local angiogenesis, epithelial tissue growth, and accelerated wound healing Lack of histological assessment of wound healing	He et al. [43]
**Vitamin E (VE) and derivates**	Murine model with MRSA-infected wounds Tocotrienols (T3s) + daptomycin, best antibacterial action vs. single treatment or placebo, better markers of wound healing (fibronectin type III expression and IL-24 mRNAs). Lack of histological assessment of wound healing	Pierpaoli et al. [44]
	Murine model with MRSA-infected wounds VE pre-treatment + daptomycin showed the highest antibacterial efficacy. VE enhancer of the antimicrobial activity of both daptomycin and tigecycline. Lack of histological assessment of wound healing	Provinciali et al. [45]
**IB-367**	Murine model with MRSA-infected wounds IB-367 + daptomycin the most effective in bactericidal activity vs. single therapies and IB-367 + teicoplanin. Lack of histological assessment of wound healing	Cirioni et al. [46]
**RNA III-inhibiting peptide** **(RIP)**	Patient with a diabetic foot ulcer RIP topical 1 mg/cc in combination with daptomycin 6 mg/kg intravenously Complete closure in 24 weeks, avoiding amputation Example of the potential of combining new topical molecules + daptomycin, increased antibacterial and improved wound healing	Lopez-Leban et al. [47]
**Dap@Au/Ag** **nanorods** **(Dap@Au/Ag NRs)**	In vitro and in vivo murine models, After exposure to 808 nm, 0.8 W/cm^2^, 40–60 s laser, Dap@Au/Ag NRs increased its antibacterial activity with a good photothermal effect. Dap@Au/Ag NRs significantly inhibit MRSA, prevent wound ulceration, promoting wound healing.	Dong et al. [48]

## Data Availability

Not applicable.

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
