# Peer review of "Role of Daptomycin in Cutaneous Wound Healing: A Narrative Review"

_antibiotics, 2022, doi:10.3390/antibiotics11070944_

Round 1
Reviewer 1 Report
The review "Role of daptomycin in wound healing: a narrative review" by Rizzetto et al., is quite well written. However illustrations and flow diagrams are entirely missing in this review which often helps to grasp the topic better. Moreover the conclusion is quite short without outlook and discussion. Several issues must be discussed in depth with appropriate references. Additionally several important and recent references are missing some of which may be relevant to include in this review. Here are some examples: Pujol et al., 2021:72(9):1517–25; Nichols et al., 2021; Schweizer et al., 2021; Tirilomis et al., 2014; Karas et al., 2020; Dong et al., 2020; Chamberlain et al., 2009; Blasco et al., 2021; Alqahtani et al., 2021 etc.
minor clarifications
Title: daptomycin: D in capital letter.
line 35: intracellular inhibition of DNA and RNA: explain how
line 40: administered parenterally: how?
line 81-82: that high doses of daptomycin (from 6 mg/kg/day up to 12
mg/kg/day) were more effective than the standard dose): won't this dose hamper the renal function of the patient?
Author Response
The review "Role of daptomycin in wound healing: a narrative review" by Rizzetto et al., is quite well written. However illustrations and flow diagrams are entirely missing in this review which often helps to grasp the topic better.
We included a table which summarize the evidence found
Moreover the conclusion is quite short without outlook and discussion. Several issues must be discussed in depth with appropriate references. Additionally several important and recent references are missing some of which may be relevant to include in this review. Here are some examples:
The main topic of this review is the role of daptomycin in cutaneous wound healing. We included some of the proposed studies with concerning the wound healing and the conclusion was improved as required. We did not include those studies without skin involvement.
Pujol et al., 2021:72(9):1517–25; Daptomycin Plus Fosfomycin Versus Daptomycin Alone for Methicillin-resistant Staphylococcus aureus Bacteremia and Endocarditis:
Alqahtani et al., 2021 etc. Influence of the minimum inhibitory concentration of daptomycin on the outcomes of Staphylococcus aureus bacteraemia,
Nichols et al., 2021; Clinical Outcomes With Definitive Treatment of Methicillin Resistant Staphylococcus aureus Bacteremia
Schweizer et al., 2021; Comparative Effectiveness of Switching to Daptomycin Versus Remaining on Vancomycin Among Patients With Methicillin-resistant Staphylococcus aureus (MRSA) Bloodstream Infections
Tirilomis et al., 2014; Daptomycin and Its Immunomodulatory Effect: Consequences for Antibiotic Treatment of Methicillin-Resistant Staphylococcus aureus Wound Infections after Heart Surgery, included
In addition, an interesting aspect of daptomycin is its possible immunomodulatory action. In an in vitro model on mononuclear cells from peripheral blood, S. aureus stimulates the production of several inflammatory cytokines, such as IL-1b, IL-6, IL-8, interferon-g (INF-g) and tumor necrosis factor alpha (TNF-a). Daptomycin suppresses the levels of these proinflammatory cytokines in vitro, independently of dose, but in vivo this does not seem to be confirmed.83 However, if VE was provided prior to wound infection with MRSA, the animals showed a significant increase in CD49b+ lymphocytes after daptomycin treatment, whereas this did not occur with daptomycin alone, suggesting its role of immunomodulator enhancer.77
Karas et al., 2020; Structure-Activity Relationships of Daptomycin Lipopeptides, included
Dong et al., 2020; The Efficacy and Safety of High-dose Daptomycin in the Treatment of Complicated Skin and Soft Tissue Infections in Asians, included
A further issue emerging from this review is the possibility of using a high dose (HD) of daptomycin for the treatment of complicated skin and soft tissue infections (cSSTIs). In a study85 of 155 Chinese patients, 108 treated with standard dose (SD) (4mg/kg) and 47 with HD (> 6mg/kg), it was shown that the group with HD achieved clinical stabilization earlier (72.34% vs 52.78%, P=0.023), with an equal distribution of adverse effects in the two groups. Form our point of view, this may be a therapeutic option that the clinician should be aware of in the treatment of cSSSIs, always considering the patient's clinical condition and his comorbidities.
Chamberlain et al., 2009; Daptomycin for the treatment of surgical site infections, included
Chamberlain et al.86 also demonstrated the efficacy of daptomycin in the treatment of chirurgic wounds infected with S. aureus, MRSA, and enterococci even when previous antibiotic therapy, including vancomycin, failed. However, evaluations on wound healing are lacking.
Blasco et al., 2021 An atomic perspective on improving daptomycin's activity included
Finally, Blasco et al. 87 recently identified a new Daptomycin analogue, kynomycin, which showed higher activity against MRSA and lower cytotoxicity than Daptomycin. This new cyclic lipopetide could represent a further option for overcoming resistance to daptomycin, but its action on wound healing still needs to be assessed.
minor clarifications
Title: daptomycin: D in capital letter
Done
line 35: intracellular inhibition of DNA and RNA: explain how
done
This occurs because adenosine tri phosphate (ATP) regeneration through oxidative phosphorylation stops when the membrane potential collapses. In addition, the loss of cellular Mg2+, which normally stabilizes ATP, should contribute to ATP depletion, which disrupt macromolecular biosynthesis, including the DNA, RNA, protein and peptidoglycan synthesis pathways.6
line 40: administered parenterally: how?
Intravenously
line 81-82: that high doses of daptomycin (from 6 mg/kg/day up to 12 mg/kg/day) were more effective than the standard dose): won't this dose hamper the renal function of the patient?
Even if this dosage is higher than what is recommended, no significant differences of endogenous creatinine clearance rate (Ccr), blood urea nitrogen (BUN), and total protein (TP) were found in patients before and 7 days after high-dose daptomycin treatment. The mean concentration of albumin on day 7 after high-dose daptomycin treatment was significantly higher than the baseline value (40.5 ± 4.6 vs 35.0 ± 6.5, respectively, P = .006)44 We recommend that the physician should always assess the risk-benefit ratio in the individual patient before considering high doses of Daptomycin.
Reviewer 2 Report
The aim of the manuscript was to evaluate the role of daptomycin in skin healing and also to describe the association with other molecules that could improve this process. The topic is relevant because daptomycin is active against Gram-positive bacteria and most skin infections are caused by staphylococci or other pyogenic cocci, such as β-haemolytic streptococci. It is important to mention that methicillin-resistant Staphylococcus aureus (MRSA) is an increasing problem worldwide.
Other comments:
- Daptomycin is commonly used as an alternative bactericidal drug against resistant bacteria. Therefore, the property of daptomycin to induce wound healing, addressed by the authors, is a positive aspect of the manuscript, as well as the analysis of the literature on its combined use with other compounds to obtain a synergistic effect and also overcome forms of antibiotic resistance. Also, there is no similar review published recently.
- The language is clear and concise, but the selected studies that were cited in the results and discussion section did not have an in-depth analysis;
- The manuscript is well organized. However, the text would be more didactic and interesting if there were at least one table (this could include a summary of clinical and experimental studies on wound healing with antibiotic use) and a figure (with the mechanisms on the immunomodulatory effects of daptomycin promoting wound healing). The manuscript can be improved with these suggestions.
- It was submitted for publication as “Perspectives”, where the article should show the developments in this field, with data from the literature in the last 3 years. However, most references (69 out of 82 articles) were published earlier. I forward some references for the authors to check if they are suitable for the manuscript:
· Dong, X. et al. Chemical Engineering Journal 2022, 432, 134061, doi:https://doi.org/10.1016/j.cej.2021.134061 - A recent study in which the authors developed a hybrid antibacterial photothermal system consisting of daptomycin linked to silver-coated gold nanorods that showed therapeutic effect on wound healing of MRSA skin infections.
· Dong, X.-m. et al. International Journal of Infectious Diseases 2020, 95, 38-43, doi:https://doi.org/10.1016/j.ijid.2020.03.060 - The authors compared the treatment of patients with skin and soft tissue infections at the standard dose and high-dose daptomycin. There was a higher clinical success rate with the high dose with no increase in adverse events.
- The authors' opinion resulting from their experience with the subject is always expressed. The conclusions are consistent and supported by the citations listed.
Minor comments:
- Line 105: Please change “Aron F. et al 59” to Popov et al 59”
- Line 111: Enterococcus is misspelled and spp. should not be in italics.
Author Response
The aim of the manuscript was to evaluate the role of daptomycin in skin healing and also to describe the association with other molecules that could improve this process. The topic is relevant because daptomycin is active against Gram-positive bacteria and most skin infections are caused by staphylococci or other pyogenic cocci, such as β-haemolytic streptococci. It is important to mention that methicillin-resistant Staphylococcus aureus (MRSA) is an increasing problem worldwide.
Other comments:
- Daptomycin is commonly used as an alternative bactericidal drug against resistant bacteria. Therefore, the property of daptomycin to induce wound healing, addressed by the authors, is a positive aspect of the manuscript, as well as the analysis of the literature on its combined use with other compounds to obtain a synergistic effect and also overcome forms of antibiotic resistance. Also, there is no similar review published recently.
- The language is clear and concise, but the selected studies that were cited in the results and discussion section did not have an in-depth analysis.
We added some aspects in the study considered
This method (Dap-PdNFs + laser group) was proved effective in vitro against S. Aureus, reducing its viability after 10 minutes of laser to 30.93% (vs. 79.76%, Dap + laser group, vs. 85.14% PdNPs + laser group). The effect on wound healing was studied on diabetic mice with S. Aureus-infected wounds. The mice were then randomly divided into five groups: saline group, saline + laser irradiation (1.75 W/cm2 for 5 min), Dap-PdNFs (topical application), Dap-PdNFs + laser irradiation. After 14 days the Dap-PdNFs + laser group showed the best reduction in wound area, 6.5%, vs laser alone group, 34.04%, vs Dap + laser group, 25%, and vs PdNPs + laser group, 17.36%. A histological examination of the main organs (heart, liver, spleen, lung and kidney) was performed, showing no alterations, thus confirming the good biocompatibility of the method. In conclusion, Dap-PdNFs irradiated with 808 nm light was effective against bacteria around the wound, prevented wound infection, promoted local angiogenesis and epithelial tissue growth, and accelerated wound healing in diabetic mice. This may be an interesting method that deserves to be evaluated with larger sample size studies, considering the topical application and the good tolerability.
In addition, an interesting aspect of daptomycin is its possible immunomodulatory action. In an in vitro model on mononuclear cells from peripheral blood, S. aureus stimulates the production of several inflammatory cytokines, such as IL-1b, IL-6, IL-8, interferon-g (INF-g) and tumor necrosis factor alpha (TNF-a). Daptomycin suppresses the levels of these proinflammatory cytokines in vitro, independently of dose, but in vivo this does not seem to be confirmed.83 However, if VE was provided prior to wound infection with MRSA, the animals showed a significant increase in CD49b+ lymphocytes after daptomycin treatment, whereas this did not occur with daptomycin alone, suggesting its role of immunomodulator enhancer.77
A further issue emerging from this review is the possibility of using a high dose (HD) of daptomycin for the treatment of complicated skin and soft tissue infections (cSSTIs). In a study85 of 155 Chinese patients, 108 treated with standard dose (SD) (4mg/kg) and 47 with HD (> 6mg/kg), it was shown that the group with HD achieved clinical stabilization earlier (72.34% vs 52.78%, P=0.023), with an equal distribution of adverse effects in the two groups. From our point of view, this may be a therapeutic option that the clinician should be aware of in the treatment of cSSSIs, always considering the patient's clinical condition and his comorbidities, but further studies with histological assessment are necessary to evaluate the impact on wound healing.
Finally, Blasco et al. 87 recently identified a new Daptomycin analogue, kynomycin, which showed higher activity against MRSA and lower cytotoxicity than Daptomycin. This new cyclic lipopeptide could represent a further option for overcoming resistance to daptomycin, but its action on wound healing still needs to be assessed.
- The manuscript is well organized. However, the text would be more didactic and interesting if there were at least one table (this could include a summary of clinical and experimental studies on wound healing with antibiotic use) and a figure (with the mechanisms on the immunomodulatory effects of daptomycin promoting wound healing). The manuscript can be improved with these suggestions.
Thank you, we added a table summarizing the main evidence as required. We did not provide a figure since we believe the table could be more easy and complete to read.
- It was submitted for publication as “Perspectives”, where the article should show the developments in this field, with data from the literature in the last 3 years. However, most references (69 out of 82 articles) were published earlier. I forward some references for the authors to check if they are suitable for the manuscript:
- Dong, X. et al. Chemical Engineering Journal 2022, 432, 134061, doi:https://doi.org/10.1016/j.cej.2021.134061 - A recent study in which the authors developed a hybrid antibacterial photothermal system consisting of daptomycin linked to silver-coated gold nanorods that showed therapeutic effect on wound healing of MRSA skin infections.
- Dong, X.-m. et al. International Journal of Infectious Diseases 2020, 95, 38-43, doi:https://doi.org/10.1016/j.ijid.2020.03.060 - The authors compared the treatment of patients with skin and soft tissue infections at the standard dose and high-dose daptomycin. There was a higher clinical success rate with the high dose with no increase in adverse events.
We included the proposed citations
- The authors' opinion resulting from their experience with the subject is always expressed. The conclusions are consistent and supported by the citations listed.
Thank you
Minor comments:
- Line 105: Please change “Aron F. et al 59” to Popov et al 59”
Done
- Line 111: Enterococcus is misspelled and spp. should not be in italics.
Done
Reviewer 3 Report
Dear Authors,
The present study ID:antibiotics-1773504 entitled "Role of daptomycin in wound healing: a narrative review" written by authors Giulio Rizzetto, Elisa Molinelli, Giulia Radi, Federico Diotallevi, Oscar Cirioni, Lucia Brescini, Andrea Giacometti, Annamaria Offidani, Oriana Simonetti.
The text is focused on an interesting topic of the role of daptomycin in wound healing, but in my opinion, the text is very brief, it does not bring a fundamentally new view of the issue (compared to other previously published studies).
Then there are errors in the keywords (redundant word, case), and the citation does not seem to me to match the style of the journal.
Author Response
Dear Authors,
The present study ID:antibiotics-1773504 entitled "Role of daptomycin in wound healing: a narrative review" written by authors Giulio Rizzetto, Elisa Molinelli, Giulia Radi, Federico Diotallevi, Oscar Cirioni, Lucia Brescini, Andrea Giacometti, Annamaria Offidani, Oriana Simonetti.
The text is focused on an interesting topic of the role of daptomycin in wound healing, but in my opinion, the text is very brief, it does not bring a fundamentally new view of the issue (compared to other previously published studies).
The aim of this review is to evaluate the role of daptomycin in the skin wound healing process, proposing our point of view on the possible association with other molecules that could improve the skin healing process.
To the best of our knowledge, there are no reviews in the literature on the role of daptomycin in wound healing. We tried to make the review brief in order to be easier to read and concise. However, we improved the results and discussion with more details.
Then there are errors in the keywords (redundant word, case), and the citation does not seem to me to match the style of the journal.
We corrected the errors we found (see revision mode) and applied the correct style for the citations.
Round 2
Reviewer 1 Report
Then change the title to ""Role of Daptomycin in cutaneous wound healing: a narrative review" instead.
I still find that a general Figure explaining the mechanisms on the immunomodulatory effects of daptomycin promoting wound healing must be included in the Manuscript as also suggested by one of the other Reviewers.
In that way, the text in too much detail provided in the TABLE could be accordingly reduced for better understanding.
Author Response
Then change the title to ""Role of Daptomycin in cutaneous wound healing: a narrative review" instead.
done
I still find that a general Figure explaining the mechanisms on the immunomodulatory effects of daptomycin promoting wound healing must be included in the Manuscript as also suggested by one of the other Reviewers.
Done, see figure 1
In that way, the text in too much detail provided in the TABLE could be accordingly reduced for better understanding.
Done, see table 1

Reviewer 3 Report
Dear Authors, Thank you for the changes made in the manuscript.
However, I present several points for correction: 1/ L.96 - correct "P" value 2/ L. 98 etc. - I don't understand why daptomycin is still listed with a capital letter in some places 3/ English language correction is needed. 4/ The citation style and method of citation in the text is definitely not according to the instructions for authors related to Antibiotics/MDPI (citations in the text in square brackets and before a period after the sentence, etc.). 5/ Table 1 seems rather confusing and unclear, it needs to be reworked, simplified, etc. In addition, there is a table with text content with different font styles, etc. I am asking for a precise revision. It would also be appropriate to avoid using bullet within table parts.
Author Response
Reviewer 2
However, I present several points for correction: 1/ L.96 - correct "P" value 2/ L.
Done
98 etc. - I don't understand why daptomycin is still listed with a capital letter in some places
Corrected
3/ English language correction is needed.
Done
4/ The citation style and method of citation in the text is definitely not according to the instructions for authors related to Antibiotics/MDPI (citations in the text in square brackets and before a period after the sentence, etc.).
done
5/ Table 1 seems rather confusing and unclear, it needs to be reworked, simplified, etc. In addition, there is a table with text content with different font styles, etc. I am asking for a precise revision. It would also be appropriate to avoid using bullet within table parts.
Done see table 1
Round 3
Reviewer 1 Report
can be accepted
Author Response
thank you
Reviewer 3 Report
Dear authors,
Thank you for the corrections. The text is ready for publication now.
Author Response
Thank you